# Computed Tomographic Findings of Dental Disease and Secondary Diseases of the Head Area in Client-Owned Domestic Rabbits (*Oryctolagus cuniculus*): 90 Cases

**DOI:** 10.3390/ani14081160

**Published:** 2024-04-11

**Authors:** Wojciech Borawski, Zdzisław Kiełbowicz, Dominika Kubiak-Nowak, Przemysław Prządka, Gerard Pasternak

**Affiliations:** 1Department and Clinic of Surgery, Faculty of Veterinary Medicine, Wroclaw University of Environmental and Life Sciences, Pl. Grunwaldzki 51, 50-366 Wroclaw, Poland; zdzislaw.kielbowicz@upwr.edu.pl (Z.K.); dominika.kubiak-nowak@upwr.edu.pl (D.K.-N.); przemyslaw.przadka@upwr.edu.pl (P.P.); 23rd Department and Clinic of Pediatrics, Immunology and Rheumatology of Developmental Age, Faculty of Medicine, Medical University of Wrocław, ul. Koszarowa 5, 51-149 Wroclaw, Poland; gerard.pasternak@umw.edu.pl

**Keywords:** pet rabbits, dental disease, CT scanning

## Abstract

**Simple Summary:**

Domestic rabbits are one of the most commonly kept pets with a hypselodontic type of dentition, possessing teeth that grow throughout the animal’s life. These animals often show signs of disease in the stomatognathic system. Dental disease can also affect the function of other organs and systems. The most common clinical signs of dental disease in domestic rabbits are the following: lack of appetite, weight loss, apathy, and difficulty chewing and swallowing food. These clinical signs are non-specific, and, therefore, a definitive diagnosis usually requires additional methods, such as an X-ray examination, a CT scan, MRI, and endoscopic examination. The most common dental disease found in this study’s animals was malocclusion secondary to abnormal clinical crown abrasion and abnormal tooth growth. In domestic rabbits, osteomyelitis is a common complication of dental abscesses. Computed tomography is an invaluable diagnostic method in the diagnosis of dental disease and secondary diseases of the head area, such as inflammation of the nasal cavities or otitis media, in pet rabbits.

**Abstract:**

Domestic rabbits have teeth that grow throughout the animal’s life and are prone to disease. Clinical signs of dental disease in domestic rabbits are non-specific, and, therefore, a definitive diagnosis usually requires additional methods. This study was carried out on a group of 105 domestic rabbits aged 3 to 9 years. In total, 90 domestic rabbits with dental disease visible on CT images and other secondary diseases of the head area qualified for this study. Malocclusion was found in 57 (63.3%). Retrograde elongation of the tooth apices in the mandible was present in 39 (43.3%), and it was present in the maxilla in 48 (53%). Clinical tooth crowns were overgrown in 39 (43%). Dental abscesses were present in 54 (63%). Secondary to the presence of a dental abscess, osteomyelitis was found in 43 (79% of the animals with a dental abscess). Dental inflammatory resorption was found in 36 (40%). Secondary to dental disease, nasal cavity inflammation was found in 18 (20%). Otitis media was present in six (6.7%). The most common dental disease found in this study’s animals was malocclusion secondary to abnormal clinical crown abrasion and abnormal tooth growth. In domestic rabbits, osteomyelitis is a common complication of dental abscesses. Computed tomography is an invaluable diagnostic method in the diagnosis of dental disease and secondary diseases of the head area, such as inflammation of the nasal cavities or otitis media, in pet rabbits.

## 1. Introduction

Small mammals such as domestic rabbits (Latin: *Oryctolagus cuniculus* f. *domesticus*) are among the most commonly kept domestic animals with hypselodont dentition, possessing teeth that grow throughout the animal’s life. This type of dentition is only found in a small group of caviomorph rodents and lagomorphs. Due to their hypselodontic dentition type, pet rabbits are frequent patients of veterinary institutions, showing signs of diseases related to the stomatognathic system. It should be noted that dental disease can also affect the functioning of other organs and systems, like the respiratory and digestive systems. Therefore, dental lesions in domestic rabbits are defined as a disease syndrome rather than a single disease entity [1]. The most common clinical signs of dental disease in these animals include the following: inappetence, weight loss, apathy, and difficulty chewing and swallowing food. It should be emphasized that the above-mentioned clinical signs are non-specific; therefore, making a definitive diagnosis of a stomatognathic disease in these animals is difficult and usually requires additional methods such as an X-ray, CT, MRI, or endoscopic examination.

Dental disease is a common disorder found in domestic rabbits. The primary cause of acquired dental disease is usually insufficient or inappropriate abrasion of the cheek teeth, which is generally caused by an inadequate diet lacking sufficient fiber content [2]. Genetic factors like hereditary brachygnathia superior have also been suggested to influence the development of acquired dental disease [3]. The high metabolic activity of odontoblasts and germinal cells of the tooth apex renders them susceptible to the adverse effects of systemic diseases and metabolic disorders, resulting in their dysfunction and the development of dental disease. In domestic rabbits, dental tissue has a high affinity for calcium, which ensures its growth and dentin mineralisation, even during long-term hypocalcaemia [4].

In acquired dental disease, inappropriate chewing and the consequent abrasion of the teeth results in a disruption of their shape, position, and structure [4]. This could result in malocclusion involving the incisors or cheek teeth. In addition, it should be noted that malocclusion of incisor teeth can cause malocclusion of cheek teeth and vice versa [3,5]. Occlusal abnormalities of the cheek teeth and abnormal growth cause elongation of the clinical crowns, uneven abrasion, and the formation of spurs or hooks, which may appear on the side of the tongue or the cheek. These lesions often cause injury to the mucosa. Abnormal growth of the clinical crowns may coexist with overgrowth of the reserve crowns and elongation of the apices of the teeth. Tooth apex elongation results in the following: irritation of nerve endings, distortion of the bone, and pressure on the apex itself, which leads to ischaemia [3].

To date, however, it has not been fully proven whether the cause of dental disease in domestic rabbits is genetic, metabolic, or due to a poor diet. It is suspected that dental diseases may comprise a combination of these three causes. However, it has been proven that any disorder causing abnormal tooth growth and abrasion will lead to the development of a dental disease syndrome [1].

The head of a rabbit with no dental disease was visualized using computed tomography and is shown in Figure 1.

### 1.1. Progressive Syndrome of Acquired Dental Disease

In rabbits, most dental diseases are part of the so-called progressive syndrome of acquired dental disease (PSADD), which affects the tooth tissues and the position and shape of the teeth [6,7]. In the course of this syndrome, Harcourt–Brown distinguished five disease stages, which are shown in Table 1 [7].

The head of a rabbit with mild dental disease (stage 2) was visualized using computed tomography and is shown in Figure 2.

### 1.2. Diagnosis of Dental Disease in Domestic Rabbits

The diagnosis of dental disease in a domestic rabbit should be based on the following: information obtained from its history, the results of the clinical examination with particular emphasis on the dental examination, and the results of additional tests. Because of the non-specific clinical signs and the impossibility of assessing the reserve crown, periodontal tissue, and adjacent bony structures via clinical and endoscopic examination, imaging diagnostic methods are required to carry out a definitive diagnosis, such as an X-ray, CT, and MRI. Abnormalities that are not detectable by clinical and endoscopic examination account for an average of 80% of all pathological processes [8]. It should be emphasised that, in domestic rabbits, a valuable adjunct to clinical examination in the diagnosis of dental disease checks for nasolacrimal duct patency [9]. In about 5–8% of cases, secondary infections of the nasolacrimal duct cause purulent discharge from the nasal cavities, and the cause cannot be detected in clinical examinations. Particularly noteworthy is the fact that more than 90% of abscesses of the head area are dental abscesses, and they are undetectable in most cases upon clinical examination [1].

### 1.3. Use of Computed Tomography in the Diagnosis of Dental Disease in Domestic Rabbits

Computed tomography is an imaging method used for diagnosing dental disease in lagomorphs and rodents and accurately assessing its severity [10]. Computed tomography allows the assessment of the full length of the teeth, the alveolar process of the maxilla and the incisor, the premolar and molar areas of the mandible, the periapical tissues, the surrounding bony structures, and after the administration of an intravenous contrast medium, the surrounding soft tissues. Furthermore, with this diagnostic method, it is possible to diagnose concomitant diseases that are directly or indirectly related to dental disease [11].

CT examination obtains better results than plain X-ray diagnoses with respect to osteolysis that is secondary to dental disease, as well as osteomyelitis [12]. In addition, examination with an intravenous contrast agent enables the diagnosis of abscesses in the head region and the determination of whether they are of dental origin; if they are, a further determination is carried out with respect to whether they are secondary to the disease and from which tooth they originated from. Three-dimensional reconstructions significantly facilitate the diagnosis of dental disease, allowing the selection of an appropriate treatment [12].

### 1.4. The Aim of this Study

This study aimed to determine the prevalence of selected dental diseases in domestic rabbits using computed tomography and to analyze the lesions of other anatomical structures associated with dental pathologies.

## 2. Materials and Methods

This study was carried out on a group of 105 domestic, client-owned rabbits aged 3 to 9 years. Rabbits were referred to the Diagnostic Imaging Department of the Faculty of Veterinary Medicine, Wroclaw University of Environmental and Life Sciences, for a head CT scan due to suspected dental disease. This study was conducted between 2018 and 2020. In total, 90 domestic rabbits with dental disease visible on CT images and other secondary diseases of the head area qualified for this study. In the remaining 15 rabbits, no features of dental diseases were found.

In accordance with Article 1, point 2.1, of the Law of 15 January 2015 on the Protection of Animals Used for Scientific or Educational Purposes [13], the research and activities carried out with client-owned animals were not subject to evaluations carried out by the 2nd Local Ethical Committee (RESOLUTION NO. 102/2017 of 21 November 2018).

### 2.1. Animal Testing Plan

The animal study was divided into three stages as follows:Qualification and preparation of the animal for CT scanning;Sedation/general anaesthesia of the animal;Computed tomography of the head with and without contrast agent.

### 2.2. Qualification and Preparation of Animals for CT Scanning

In all animals, a history and clinical examination was carried out prior to the head CT scan, with a particular focus on the stomatognathic system.

In addition, blood tests (hematological tests—RBC, HGB, HCT, MCV, MCH, MCHC, WBC, leucogram, and PLT; biochemical tests—AST, ALT, ALP, urea, creatinine, total protein, albumin, DGGR, lipase, and glucose) were performed in all animals. Animals were not fasted prior to sedation/general anaesthesia. Each animal had a peripheral venous access catheter inserted into the marginal ear vein.

### 2.3. Sedation/General Anaesthesia of the Animals

The following anaesthesia regimens were used in animals qualified for CT scans:4.Sedation: Medetomidine (Cepetor, 1 mg/mL, ScanVet, Gniezno, Poland) at a dose of 0.1 mg/kg b.w. and ketamine (Bioketan 100 mg/mL, Vetoquinol, Gorzów Wielkopolski, Poland) at a dose of 5 mg/kg b.w. given in one intramuscular injection;5.General anaesthesia: In cases where sedation was insufficient for CT scanning, animals were administered general anaesthesia with propofol (Propofol-Lipuro, 10 mg/mL, Braun, Melsungen, Germany) at an initial dose of 1–2 mg/kg b.w. administered in a single intravenous injection. Subsequent doses of propofol were administered as needed.

### 2.4. Computed Tomography of the Head

Computed tomography of the head was performed using 16-slice Siemens Somatom Emotion computed tomography. Animals were placed on a sponge positioner, in a sternal position, in line with the long axis of the CT table, head facing the gantry. In order to maximise the exposure of the animal’s head, the positioner was placed underneath it. Scanning was performed in the long axis of the head, first in the cranial and then in the caudal direction. After the head overview CT scan, each animal was injected intravenously with the contrast agent Iomeprol (Iomeron 350 mg iodine/mL, Bracco, Konstanz, Germany) at a dose of 700 mg/kg body weight. The head CT scan was performed using the following exposure parameters: 60 mAs and 130 kV, with a sliding factor of 0.75. The time per scan was approximately 90 s. Cross-sections of 0.6 mm thickness were obtained. Images of the cross-sections were obtained using a bone filter with W 1400, C 300, and soft tissue. Head images were obtained using Siemens syngoMMWP software, syngo.via.vb20b. In addition, the multiplanar reconstruction image function in sagittal, dorsal, and transverse sections and the 3D image function were used.

CT-derived images were analyzed in multiplanar reconstructions in sagittal, coronal, and transverse sections, as well as in 3D reconstruction, where the following elements were assessed:The mutual positioning of the mandibular and maxillary incisors and the mandibular and maxillary cheek teeth in relation to each other in order to assess the presence of malocclusion;Reserve crowns of incisors and cheek teeth for retrograde apical elongation;The presence of mandibular bone deformities secondary to the presence of the retrograde elongation of the apices of the incisor and cheek teeth;The presence of the deformation occurrences of the maxilla bone and the destruction of the orbital bone wall, secondary to the presence of retrograde elongation of the apices of the incisor and cheek teeth;Clinical crowns of incisors and cheek teeth for overgrowth;Clinical and reserve crowns of incisors and cheek teeth for inflammatory resorption;Soft tissues for the presence of dental abscesses;Mandibular and maxillary bone for osteomyelitis secondary to the presence of an abscess of dental origin;Nasal cavities for inflammation secondary to dental disease;Tympanic bulla for otitis media secondary to dental disease.

## 3. Results

### 3.1. Prevalence of Malocclusion

On the basis of a head CT scan of 90 domestic rabbits, 57 were diagnosed with malocclusion. The number of animals with malocclusion of the cheek teeth was 54, while that of the incisors was 36. Among the animals with malocclusion of the cheek teeth, in 24 cases, malocclusion occurred unilaterally, and in 30 cases, it occurred bilaterally. In contrast, all animals with occlusion defects at the incisor tooth level had bilateral occlusion. Among the animals with occlusal malocclusion, in twenty-one cases, the abnormality only affected the cheek teeth, and only the incisor teeth were affected in three cases. On the other hand, in 33 animals, malocclusion at the level of the cheek teeth coexisted with malocclusion of the incisor teeth. The prevalence of malocclusion in the group of domestic rabbits studied is shown in Table 2.

### 3.2. Prevalence of Retrograde Apical Elongation in the Mandible in Domestic Rabbits

In the head CT, retrograde elongation of maxillary tooth apices was present in 39 domestic rabbits. Among these animals, they were present bilaterally in thirty-six animals and unilaterally in three animals. In contrast, retrograde elongation of the apices of incisors was found in three animals. Retrograde elongation of the apices of premolars occurred in 33 animals, and with respect to molars, it occurred in 36 animals. The apices of premolars overgrew retrospectively on their own in three animals, and with respect to molars, this occurred in six animals. In contrast, the apices of both premolars and molars overgrew retrospectively in 30 animals. In addition, 21 animals developed mandibular bone deformities as a result of the retrograde elongation of the apices. Of these, mandibular deformities occurred bilaterally in eighteen animals and unilaterally in three animals. Mandibular deformity at the level of the premolars and molars was present in 18 animals, 15 of which coexisted with each other. The prevalence of retrograde elongation of the mandibular apices in domestic rabbits is shown in Table 3.

### 3.3. Prevalence of Retrograde Apical Elongation in the Maxilla and Bone Deformities Secondary to It

Based on head CT scans, retrograde elongation of the maxillary tooth apices was found in forty-eight domestic rabbits, of which thirty-nine exhibited bilateral elongation and nine exhibited unilateral elongation. In addition, nine animals exhibited retrograde elongation of both the apices of the cheek teeth and the incisors. None of the animals had a retrograde elongation of the apexes of the incisors alone. Retrograde elongation of the apices of the premolars occurred in 45 animals and 36 animals with respect to the molars. The apices of the premolars overgrew independently in twelve cases, and in three cases, they overgrew with respect to the molars. On the other hand, the apexes of both the premolars and molars were retrograded in 33 animals. In addition, 15 animals exhibited a deformation of the maxillary body as a result of the retrograde elongation of the tooth apexes, and 12 animals exhibited orbital bone wall destruction. The retrograde elongation incidences of the maxillary tooth apexes and secondary bone deformities in domestic rabbits are shown in Table 4.

### 3.4. Prevalence of the Clinical Crown Overgrowth of Teeth in the Maxilla and Mandible

With respect to head computed tomography, overgrowth of the clinical crowns of the cheek teeth was present in 39 domestic rabbits. Overgrowth of the clinical crowns of the cheek teeth in the mandible was present in twenty-four animals, of which nine animals exhibited unilateral overgrown and fifteen animals exhibited bilateral overgrowth. In addition, overgrowth of the clinical crowns of the cheek teeth in the maxilla was found in 39 cases, unilaterally in 21 animals and bilaterally in 18 animals. The simultaneous overgrowth of the clinical crowns of the cheek teeth in the mandible and maxilla was found in 24 animals. The clinical crown overgrowth of the incisors was found in 15 cases, all of which occurred bilaterally in both the mandible and maxilla, coexisting with cheek tooth overgrowth. The prevalence of clinical crown overgrowth of teeth in the mandible and maxilla in domestic rabbits is shown in Table 5.

### 3.5. Incidence of Dental Abscesses and Secondary Osteomyelitis

On the basis of CT scans, dental abscesses were found in 54 cases. In 42 animals, the dental abscesses originated from a single tooth, and they originated from more than one tooth in 11 animals. Of these, in 28 cases, they were single abscesses, and in 25, they were multiple. In 34 animals, the dental abscesses originated from maxillary teeth, and in 31 animals, they originated from mandibular teeth. In nine animals, dental abscesses originating from both mandibular and maxillary teeth were observed. Dental abscesses originating from incisor teeth were found in five animals; from premolar teeth, twenty-three animals; and from molars, forty animals. Secondary to these, osteomyelitis was found in 43 cases, while retrobulbar abscesses were found in 18 cases. In contrast, 13 cases developed a fistula in the ocular region. The incidences of dental abscesses and secondary osteomyelitis in domestic rabbits are shown in Table 6. The head and local soft tissues of a rabbit with multiple dental abscesses are shown in Figure 3.

### 3.6. Prevalence of Inflammatory Teeth Resorption

Upon head CT scanning, inflammatory tooth resorption was found in 36 domestic rabbits. In twenty-seven animals, it affected the cheek teeth, with twenty-four cases being unilateral and three being bilateral. Among these animals, the inflammatory resorption of the cheek teeth of the mandible was found in all cases, inflammatory resorption of the cheek teeth of the maxilla was found in 21 animals, and inflammatory resorption of the cheek teeth of both the maxilla and mandible was found in 18 cases. In contrast, inflammatory resorption of incisor teeth was present in 15 cases. Nine animals had affected incisors in the mandible, nine animals had affected incisors in the maxilla, and three animals had affected incisors in both the mandible and maxilla. In four animals, the resorption of incisors was unilateral, and in one animal, it was bilateral. In six animals, the inflammatory resorption of both incisor and cheek teeth was observed. The prevalence of inflammatory teeth resorption in domestic rabbits is shown in Table 7. The head of a rabbit with inflammatory cheek teeth resorption is shown in Figure 4.

### 3.7. Prevalence, Secondary to Dental Disease, of Nasal Cavity Inflammation and Otitis Media in Domestic Rabbits

In domestic rabbits, nasal cavity inflammation secondary to dental disease was found in eighteen cases (20%), of which nine were a result of incisor tooth disease and nine were a result of cheek tooth disease. It occurred unilaterally in twelve animals and bilaterally in six. Otitis media was found in six animals (6.7%), and in all cases, it was unilateral. The head of a rabbit with secondary dental disease nasal cavity inflammation is shown in Figure 5, and the head of a rabbit with secondary dental disease middle ear inflammation is shown in Figure 6.

## 4. Discussion

This study, which aimed to determine the prevalence of selected dental diseases in domestic rabbits using computed tomography, is an important addition to the knowledge of the course and development of stomatognathic disease in these animals. Most dental lesions in domestic rabbits cannot be detected without the use of advanced imaging methods.

Malocclusion was present in 57 (63%) of the 90 rabbits examined. The occlusion defects on incisor teeth were always bilateral, and in three animals, the occlusion defect on the incisor teeth did not coexist with an occlusion defect on the cheek teeth. This was most likely due to a genetic defect in rabbits, manifested in a maxilla that is too short in relation to the mandible, which, as described by Böhmer and Kostlin in their study, results in the abnormal abrasion of incisors and thus the excessive overgrowth of the clinical crowns [14].

In 33 (58%) of the rabbits with malocclusion, occlusal abnormalities of incisor teeth coexisted with those of cheek teeth, confirming Mans and Jekl’s hypothesis that the occlusal abnormalities of cheek teeth could cause occlusal abnormalities of incisor teeth and vice versa [3]. In our study, there was also no difference in the prevalence of unilateral or bilateral cheek teeth malocclusion in domestic rabbits.

According to the literature, there are two theories that may explain the development of dental disease in pet rabbits [1,7]. The first one assumes that, as a result of insufficient teeth abrasion during the chewing of food, the clinical crowns of the mandibular and maxillary teeth start to overgrow, and by exerting increasing pressure on themselves, they affect the apexes, which begin to grow backwards, deforming the surrounding bone structures.

According to the second theory, the first stage of dental disease is not the overgrowth of the clinical crowns of the teeth but the weakening of the bone supporting the apex of the tooth. This occurs as a result of calcium and/or vitamin D deficiencies (often with a concomitant phosphorus excess). This results in retrograde apical elongation, bone deformity, and at a later stage, abnormal tooth growth [1,7,15].

In our study, retrograde apical elongation not only co-occurred with clinical crown overgrowth but also occurred alone in some animals. Similarly, clinical crown overgrowth occurred alone in some animals and co-occurred with retrograde apical elongation in others. Consequently, our study does not allow us to refute or confirm any of the previously cited theories of the development of dental disease in rabbits.

The retrograde apical elongation of teeth in the mandible occurred in 39 (43%) domestic rabbits, and it occurred in the maxilla in 48 (53%) domestic rabbits. Most occurred bilaterally and coexisted with each other, which is understandable because when the teeth on one side are overgrown, the chewing process of food is completely disrupted; consequently, further pathologies develop. Isolated cases of retrograde apical elongation on one side of the animal are concerning in the initial phase of dental disease, which, if not detected in the examination at such an early stage, would in time lead to disturbances in the growth of the teeth of the opposite side. Retrograde elongation of both the premolar and molar apices was observed, but retrograde elongation of the apices of the mandibular incisors was observed in only three animals. Elongation of the incisor teeth in the maxilla occurred in nine animals, and it always coexisted with the elongation of cheek teeth. This confirms the view of Derbaudreghien et al., who noted that incisor disease is more likely to coexist with cheek teeth disease, with molars being the most commonly affected, which is reflected in our study [16].

Overgrowth of the clinical crowns of teeth occurred in 39 (43%) of the domestic rabbits in our series. Furthermore, overgrowth of the clinical crowns of the incisors was more frequent in the animals in this study than the retrograde elongation of the apices of the incisors.

A very common complication of dental disease in small mammals with hypselodontic dentition is dental abscesses and secondary osteomyelitis. The incidence of dentigerous abscesses in domestic rabbits is, according to the literature, more frequent than, for example, in guinea pigs [17]. This is most likely due to the fact that rabbits have very weak alveolar dental ligaments, and one of the consequences of teeth overgrowth is the formation of gaps between them. Bacteria enter these gaps, resulting in the development of an abscess [18]. The presence of dental abscesses is associated with moderate to very extensive life-threatening osteomyelitis [4,19]. Even though root elongation, crown deformities, malocclusion, dental spurs, and food impaction between teeth may predispose them to dental abscesses, the pathophysiologic process itself is not fully clear [12,20].

Our study confirms the previous observations of other authors that dental abscesses are a common complication of dental disease in domestic rabbits. In our study, they were found in 54 (63%) animals with dental disease. The abscesses originated from cheek teeth in the majority of cases, with only five cases originating from incisor teeth. In our series, dental abscesses caused osteomyelitis in 43 (79%) cases of animals with dental abscesses. Furthermore, it is noteworthy that multiple abscesses and those located behind the eyeball were observed (eighteen cases), some causing a fistula in the orbital region (thirteen cases). The anatomical structure of the rabbit skull and the shape and position of the reserve crowns of the teeth in the maxillary alveoli predispose the animal to the formation of abscesses precisely in the orbital region, as described. In rabbits, the anatomical positioning of the roots of premolar and molar teeth is situated cranially in relation to the eyeballs. This anatomical fact may account for the elevated frequency of retrobulbar abscess, exophthalmos, and presence of fistula in the orbital region in rabbits, where the deformation of the retrobulbar space could more easily cause eyeball dislocation, as described by Petrini et al. in their study [21].

Following inflammatory conditions, such as periodontitis and abscesses, the loss of tooth substance can occur in the form of inflammatory teeth resorption. It has been proven that a decrease in the density of the apex, osteolysis of the bone near the apex, and elongation of the apex itself are directly related to the development of inflammatory tooth resorption in rabbits [22]. Progressive inflammatory teeth resorption can cause tooth fragmentation, making subsequent treatment more difficult. The assessment of teeth with CT scans prior to extraction can alert clinicians to the presence of fragments hindering extraction, thus preventing iatrogenic mandibular fractures during an overly aggressive surgical intervention at a site of bone weakness. Inflammatory resorption in domestic rabbits was observed in thirty-six (40%) animals, six of which exhibited inflammatory resorption that affected both incisor and cheek teeth; in the remaining cases, only cheek teeth were affected. The areas of resorption were found in both the mandible and the maxilla. We speculate that the frequent occurrence of such lesions at the cheek teeth level in rabbits is a result of their previously described and confirmed dental abscess formation tendency, and the risk of infections at the site of interdental spaces is increased due to malocclusion.

Due to the topography of the domestic rabbit dentition, the upper incisors run adjacent to the nasal cavities and paranasal sinuses, rendering secondary nasal cavity and paranasal sinus inflammation a common complication of incisor disease. In our series, inflammation of the nasal cavities was found in eighteen (20%) animals, nine of which were due to advanced elongation and periapical abscess of the first premolar tooth, while the other nine were due to incisor tooth disease. This shows that secondary rhinitis in this species can occur not only due to dental disease of the upper incisors but also due to dental disease in the first cheek teeth of the maxilla. This fact deserves special attention in the case of nasolacrimal duct obstruction, which can occur secondarily to apex elongation or the inflammation of the nasal cavities resulting from dental disease. This obstruction is often incorrectly described in the literature in the context of incisor tooth disease [23].

Otitis media were also found in six (7%) cases in our series, which probably resulted in secondary dental disease via an ascending route through the pharyngeal opening of the auditory tube [24]. All six cases of otitis media were unilateral and corresponded to the side on which the dental disease was more advanced.

These findings show how useful CT is, not only in diagnosing subtle changes in the teeth themselves but also in identifying complications of dental disease affecting other structures of the head.

## 5. Conclusions

The most common dental disease found in the study animals was malocclusion secondary to abnormal clinical crown abrasion and abnormal tooth growth. In domestic rabbits, osteomyelitis is a common complication of dental abscesses. CT scanning is an invaluable diagnostic method in the diagnosis of dental diseases and secondary diseases of the head area, such as inflammation of the nasal cavities or otitis media, in pet rabbits.

## Figures and Tables

**Figure 1 animals-14-01160-f001:**
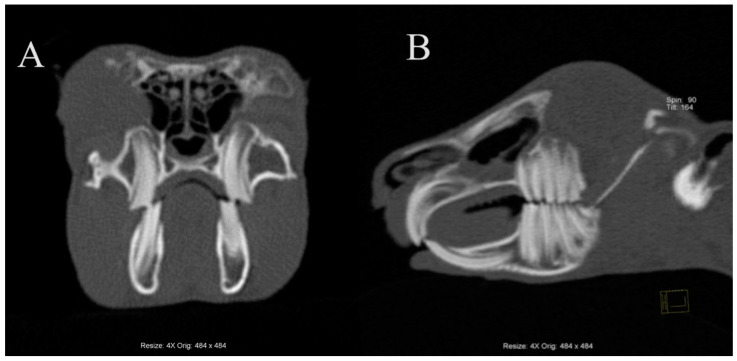
Transverse plane (**A**) and sagittal plane (**B**) reconstructed CT images showing the head of a rabbit with no signs of dental disease. Note the following: the mandible has a smooth ventral border; tooth roots are of optimal length; interdigitation of the occlusal surfaces of the cheek teeth; and parallel smooth linear pattern of the premolar and molar teeth.

**Figure 2 animals-14-01160-f002:**
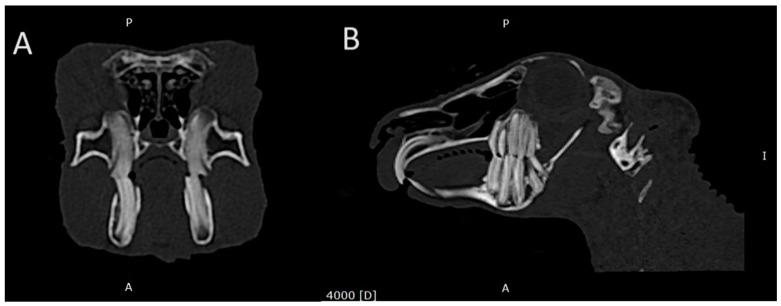
Transverse plane (**A**) and sagittal plane (**B**) reconstructed CT images showing the head of a rabbit with stage 2 dental disease. Note the following: elongation of the reserve crown and then the clinical crown. Change in shape, position, and structure of teeth; uneven occlusal surface.

**Figure 3 animals-14-01160-f003:**
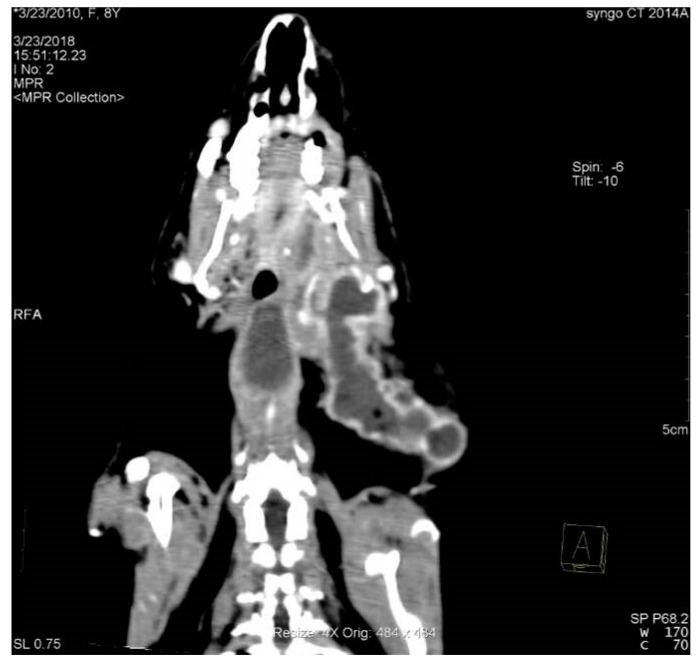
Coronal plane reconstructed CT image depicting periapical abscesses extending into local soft tissues in a rabbit. Image obtained after IV contrast medium administration viewed with soft tissue window settings. Note the capsular ring enhancement with the contrast of the multilobular abscess extending throughout the regional soft tissues up to the shoulder.

**Figure 4 animals-14-01160-f004:**
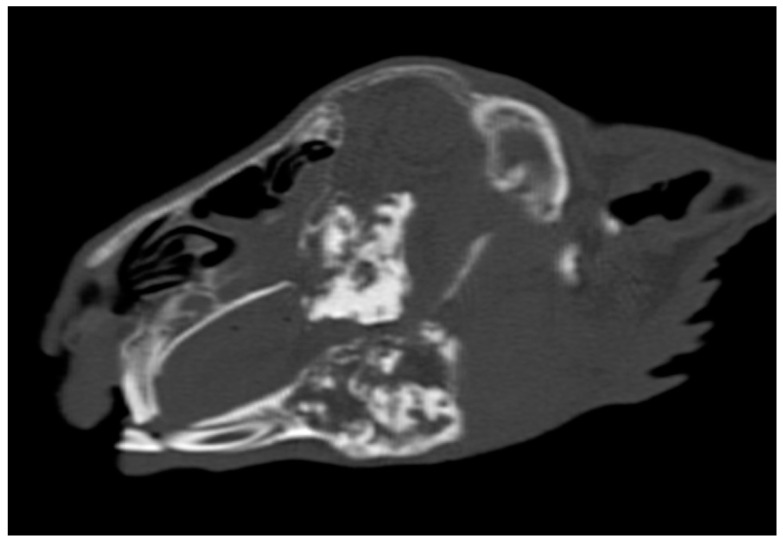
Sagittal plane reconstructed CT images showing the head of a rabbit with inflammatory cheek teeth resorption. Note that the integrity of the cheek teeth and structure were lost.

**Figure 5 animals-14-01160-f005:**
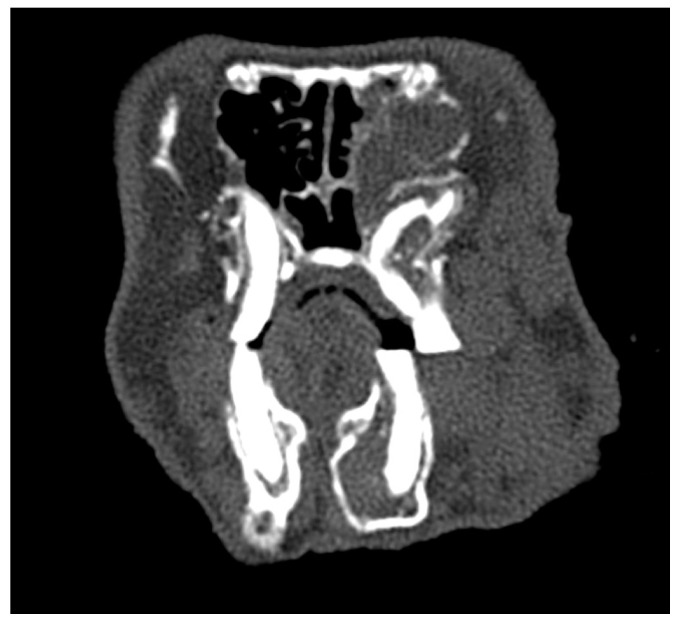
Transverse plane reconstructed CT images showing nasal cavity and sinus inflammation secondary to regional dental disease. Note the effacement and lysis due to elongated tooth apices and possible periapical abscess formation.

**Figure 6 animals-14-01160-f006:**
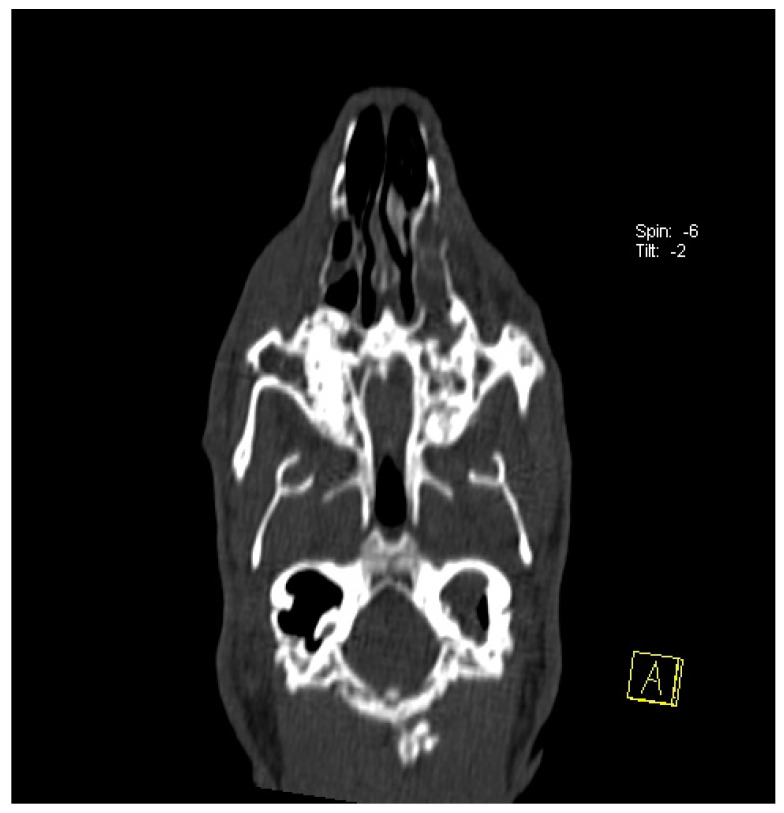
Coronal plane reconstructed CT image depicting secondary dental disease otitis media. Note the soft tissue density in the middle ear cavity and cheek teeth inflammatory resorption on the same side.

**Table 1 animals-14-01160-t001:** Degrees of lesion progression of progressive acquired dental disease syndrome according to Harcourt–Brown (2016).

Degree	Characteristics
1	No features of the disease.
2	Elongation of the reserve crown and then the clinical crown. Change in shape, position, and structure of teeth; uneven occlusal surface.
3	Acquired malocclusion. On the surface of the second and third cheek teeth of the mandible, enamel manufacturing changes in the form of spurs. Overgrowth of the first and second maxillary cheek teeth towards the oral vestibule. Possible enamel loss and dentin formation abnormalities.
4	Tooth growth inhibition.
5	Advanced, end-stage dental disease—the crowns of the teeth disintegrate or break below the gum line.

**Table 2 animals-14-01160-t002:** Prevalence of malocclusion in a group of pet rabbits.

Number of Animals	Number of Animals with Malocclusion (*n =* 90)			
**90**	57 (63.3%)			
	**Cheek teeth** **(*n =* 57)**		**Incisor teeth** **(*n =* 57)**	
	54 (95%)		36 (63%)	
	**Single-sided** **(*n =* 54)**	**Both sides** **(*n =* 54)**	**Single-sided** **(*n =* 36)**	**Both sides** **(*n =* 36)**
	24 (44%)	30 (55%)	0	36 (100%)
	**Number of animals with malocclusion of cheek teeth only** **(*n =* 57)**		**Number of animals with malocclusion of incisor teeth only** **(*n =* 57)**	
	21 (37%)		3 (5%)	
	**Number of animals with malocclusion of both incisor and cheek teeth** **(*n =* 57)**			
	33 (58%)			

**Table 3 animals-14-01160-t003:** Prevalence of retrograde apical elongation in the mandible of domestic rabbits.

Number of Animals	Number of Animals with Elongated Apices (*n* = 90)		Number of Animals with Elongation of Incisor Apices (*n =* 39)	Number of Animals with Elongation of Cheek Teeth Apices (*n =* 39)			Number of Animals with Mandibular Bone Deformity (*n =* 39)				
**90**	**39 (43%)**		**3 (1%)**	**39 (100%)**			**21 (54%)**				
	**Single-sided** **(*n =* 39)**	**Both sides** **(*n =* 39)**		**Premolar teeth** **(*n =* 39)**		**Molars** **(*n =* 39)**	**Single-sided** **(*n =* 21)**	**Both sides** **(*n =* 21)**	**At the level of premolar teeth** **(*n =* 21)**		**At molar level** **(*n =* 21)**
	3 (8%)	36 (92%)		33 (85%)		36 (92%)	3(14%)	18 (86%)	18 (86%)		18 (86%)
				**Premolar teeth only** **(*n =* 13)** **(*n =* 39)**	**Molar teeth only** **(*n =* 13)** **(*n =* 39)**	**Both premolar and molar teeth** **(*n =* 13)** **(*n =* 39)**			**Only at the level of premolar teeth** **(*n =* 7)** **(*n =* 21)**	**Only at the molar level** **(*n =* 7)** **(*n =* 21)**	**At the level of both premolar and molar teeth** **(*n =* 7)** **(*n =* 21)**
				3 (8%)	6 (15%)	30 (77%)			3 (14%)	3 (14%)	15 (71%)

**Table 4 animals-14-01160-t004:** Prevalence of the retrograde elongation of maxillary teeth apices and secondary bone deformities in domestic rabbits.

Number of Animals	Number of Animals with Apex Elongation (*n =* 90)								
**90**	**48 (53%)**								
	**Single-sided** **(*n =* 48)**	**Both sides** **(*n =* 48)**	**Elongation of both incisors and cheek teeth apices** **(*n =* 48)**	**Elongation of incisor teeth only** **(*n =* 48)**	**Elongation of cheek teeth only** **(*n =* 48)**	**Elongation of premolar teeth** **(*n =* 48)**	**Elongation of molar teeth** **(*n =* 48)**	**Deformity of the body of the mandible** **(*n =* 48)**	**Deformity of the orbital bone wall** **(*n =* 48)**
						45 (94%)	36 (75%)	15 (31%)	12 (25%)
	9 (19%)	39 (81%)	9 (19%)	0	48 (100%)	**Elongation of premolar teeth only** **(*n =* 48)**	**Elongation of molar teeth only** **(*n =* 48)**		
						12 (25%)	3 (6%)		
						**Elongation of both premolar and molar teeth** **(*n =* 48)**			
						33 (69%)			

**Table 5 animals-14-01160-t005:** Prevalence of clinical crown overgrowth.

Number of Animals	Number of Animals with Overgrowth (*n =* 90)							
**90**	**39 (43%)**							
	**Overgrowth of incisor teeth** **(*n =* 39)**				**Overgrowth of cheek teeth** **(*n =* 39)**			
	15 (38%)				39 (100%)			
	**Single-sided** **(*n =* 15)**	**Both sides** **(*n =* 15)**	**Mandible** **(*n =* 15)**	**Maxilla** **(*n =* 15)**	**Mandible** **(*n =* 39)**		**Maxilla** **(*n =* 39)**	
	0	15 (100%)	15 (100%)	15 (100%)	24 (61%)		39 (100%)	
			**Overgrowth in both the mandible and maxilla** **(*n =* 15)**		**Single-sided** **(*n =* 24)**	**Both sides** **(*n =* 24)**	**Single-sided** **(*n =* 39)**	**Both sides** **(*n =* 39)**
			15 (100%)		9 (37%)	15 (62%)	21 (54%)	18 (46%)
					**Overgrowth of cheek teeth in both the mandible and maxilla (*n =* 13)** **(*n =* 39)**			
					24 (61%)			
**Overgrowth of both incisor and cheek teeth (*n =* 13)** **(*n =* 39)**								
15 (38%)								

**Table 6 animals-14-01160-t006:** Incidence of dental abscesses and secondary osteomyelitis in domestic rabbits.

Number of Animals	Number of Animals with Dental Abscesses (*n =* 90)						
**90**	**54 (63%)**						
	**Incisor teeth** **(*n =* 54)**	**Premolar teeth** **(*n =* 54)**	**Molars** **(*n =* 54)**	**Maxilla** **(*n =* 54)**	**Mandible** **(*n =* 54)**	**Osteomyelitis** **(*n =* 54)**	**Number of retrobulbar abscesses** **(*n =* 54)**
	5 (10%)	23 (42%)	40 (74%)	34 (63%)	31 (58%)	43 (79%)	18 (37%)
	**Abscesses originating from 1 tooth** **(*n =* 54)**	**Abscesses originating from >1 tooth** **(*n =* 54)**		**Both the maxilla and mandible** **(*n =* 54)**			**Presence of a fistula** **(*n =* 18)**
	42 (78%)	11 (20%)		9 (16%)			13 (71%)
	**Single abscess** **(*n =* 54)**			**Multiple abscesses** **(*n =* 54)**			
	28 (52%)			25 (46%)			

**Table 7 animals-14-01160-t007:** Prevalence of inflammatory teeth resorption in domestic rabbits.

Number of Animals	Number of Animals with Inflammatory Teeth Resorption (*n =* 90)							
**90**	**36 (40%)**							
	**Cheek teeth** **(*n =* 36)**				**Incisor teeth** **(*n =* 36)**			
	**27 (75%)**				**15 (42%)**			
	**Single-sided** **(*n =* 27)**	**Both sides** **(*n =* 27)**	**Mandible** **(*n =* 27)**	**Maxilla** **(*n =* 27)**	**Single-sided** **(*n =* 15)**	**Both sides** **(*n =* 15)**	**Mandible** **(*n =* 15)**	**Maxilla** **(*n =* 15)**
	24 (89%)	3 (11%)	27 (100%)	21 (78%)	12 (80%)	3 (20%)	(60%)	(60%)
			**In both the mandible and the maxilla** **(*n =* 27)**				**In both the mandible and the maxilla** **(*n =* 15)**	
			18 (67%)				3 (20%)	
**Number of animals with inflammatory resorption of both cheek and incisor teeth** **(*n =* 36)**								
6 (17%)								

## Data Availability

The data generated in this study are in this article.

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
