# Peer review of "Computed Tomographic Findings of Dental Disease and Secondary Diseases of the Head Area in Client-Owned Domestic Rabbits (Oryctolagus cuniculus): 90 Cases"

_animals, 2024, doi:10.3390/ani14081160_

Round 1
Reviewer 1 Report
Comments and Suggestions for Authors
Dear authors,
There are already articles that describe the same topic as your article. I understand the time you spent on the study. However, there are other articles already published on this topic, and the number of animals in comparison with other studies is not sufficient to recommend this article for publication. Please see ...
Artiles, C.A.; Sanchez-Migallon Guzman, D.; Beaufrere, H.; Phillips, K.L. Computed tomographic findings of dental disease in domestic rabbits (Oryctolagus cuniculus): 100 cases (2009–2017). J. Am. Vet. Med. Assoc. 2020, 257, 313–327
Petrini D, Puccinelli C, Citi S, Del Chicca F. Computed Tomographic Findings Secondary to Dental Pathologies: Comparison between Rabbits and Guinea Pigs. Vet Sci. 2023 Dec 14;10(12):705. doi: 10.3390/vetsci10120705. PMID: 38133256; PMCID: PMC10747827.
So I cannot recommend this article for publication.
With best regards
Comments on the Quality of English Languageclinical crown overgrowth - use"elongation", and many other minor to moderate English spelling errors
Author Response
Thank you verry much for your time and valuable feedback.
I hope my next articles will meet your expectations.
your faithfully

Reviewer 2 Report
Comments and Suggestions for Authors
Line 31. In the abstract you forgot to mention that the 90 rabbits were presented to the CT department for dental problems. Therefore, the percentages you cite are not percentages of the whole population but rather of a group of dental patients. You do mention that fact in the materials and results but you should do so in the abstract as well.
line 50. In most rodents only the incisors are elodont. Only in a small group of caviomorph rodents is the dentition similar to the one of rabbits.
Line 72. It is rare for the incisors to be the primary cause of elongation.
Line 74. Do you have reference for this statement? Cheek teeth in rabbits are always maloccluded (the mandible is narrower than the maxilla). Malocclusion in the incisors is not so common.
Line 80. Apices do not really elongate but they move apically
Line 100. Which jaw? the maxilla?
Line114: I would place cheek teeth before incisors.
Line 185. Better diagnosis compared to what?
Line 310. hypertrophy in which direction. I think overgrowth is a better term
Comments on the Quality of English LanguageLine 71 change tounge to tongue
Line 101. use cross bite instead of abnormal position
Line 107. Replace with by secondary to
Line 112. change tounge to tongue
Line 156. change Horcout to Harcourt
Lines 278, 279, 281, 292, 295, 297, 299 change apexes to apices
Line 317. add comas , secondary to them,
Line 380. jaw change to maxilla
Author Response
First of all, I would like to thank you for your constructive comments. As you suggested, I added information about the number of examinated rabbits to the abstract and I made the remaining changes. As suggested by second reviewer, I significantly shortened the introduction. I also decided to ask for help english language editing service. I hope that the changed article will meet your expectations. Thank you again for your time and invaluable help.

Reviewer 3 Report
Comments and Suggestions for Authors
Dear Authors
I am perplexed by your paper. You have such wonderful data but started writing a Review paper. Your Introduction is far too long for a research paper and sadly also poorly written. I have made several comments and suggestions by using comment stickers on the PDF. Your scientific English also needs work. As a non-English speaking person myself I understand the challenges and suggest you get a native English scientist to help you. As a reasearch paper you will need to shorten your Introduction substantially. It is important that you get to what you did in the paper and not rehash all the literature. I curtailed my review towards the end of the Introduction and realised this problem with the paper (Review vs research). I told the Editor I would be happy to review your paper again (and look forward to it), but the style and English needs to be corrected first.

Comments on the Quality of English LanguageThe English, and in particular scientific English is generally poor. Concepts like enamel manufacturing is just not applicable. Authors would use words like often or very often but it does not add anything and is confusing. I am concerned that you use definitions which is apperent it does not have the meaning you think or want it to have (eg clinical crown hypertrophy)..
Author Response
First of all, I would like to thank you for your constructive comments. They were very important to me as a beginner in publishing my own research.
As you suggested I significantly shortened the introduction and decided to ask for help english language editing service. I also added an "image 2" showing the head of a rabbit with mild dental disease which can be compared to healthy rabbit in "image1".
I hope that the changed article will meet your expectations.
Thank you again for your time and all valuable help.
Yours faithfully,
Wojciech Borawski
